# Predictors of Tetanus Vaccine Uptake among Pregnant Women in Khartoum State, Sudan: A Hospital-Based Cross-Sectional Study

**DOI:** 10.3390/vaccines11071268

**Published:** 2023-07-21

**Authors:** Zienab A. Ibrahim, Majdi M. Sabahelzain, Yasir Ahmed Mohammed Elhadi, Ombeva Oliver Malande, Suad Babiker

**Affiliations:** 1Federal Ministry of Health, Khartoum P.O. Box 303, Sudan; zienabahmedibrahim@gmail.com; 2Nutrition and Health Center for Training and Research, Ahfad University for Women, Omdurman P.O. Box 167, Sudan; 3Public Health Department, Sudanese Medical Research Association, Khartoum P.O. Box 303, Sudan; hiph.yelhadi@alexu.edu.eg; 4Department of Child Health and Paediatrics, Egerton University, Nakuru P.O. Box 3366-20100, Kenya; ombevaom@gmail.com; 5East Africa Centre for Vaccines and Immunization (ECAVI), Namela House, Kampala P.O. Box 3040, Uganda; 6Department of Paediatrics & Child Health, Makerere University, Kampala P.O. Box 3040, Uganda; 7School of Medicine, Ahfad University for Women, Omdurman P.O. Box 167, Sudan

**Keywords:** tetanus toxoid vaccine, vaccine uptake, tetanus, pregnant women, Sudan

## Abstract

Tetanus toxoid (TT) vaccination during pregnancy has been proven as an effective preventative measure to reduce the incidence of maternal and neonatal morbidity and mortality worldwide. This study aimed to assess the determinants of TT vaccine uptake among pregnant women at two public maternity specialized hospitals in Sudan. A hospital-based cross-sectional study was conducted at two public hospitals, Omdurman Maternity Hospital and Al Saudi Hospital in Omdurman, Khartoum State, in Sudan from February to April 2020. Logistic regression analysis was carried out to identify factors associated with receiving three or more doses of the TT vaccine among pregnant women, presented as odds ratios, with *p*-values < 0.05 considered significant (at a 95% confidence interval). The study recruited 350 pregnant women, with 313 participants included in the analysis. This study found that only 40% of the pregnant women had received three doses or more of the TT vaccine. Pregnant women who attended Al Saudi Hospital were less likely to be vaccinated with the recommended dose of the TT vaccine in districts at high risk (received ≥3 doses) compared to those who attended Omdurman Hospital [OR = 0.49 (95% C.I. 0.29–0.82), *p*-value < 0.05]. Furthermore, the number of children at home was a significant predictor of the mothers’ immunization status as those with five children or more were ten times more likely to be vaccinated with three doses or more [OR = 10.54 (95% C.I. 4.30–25.86), *p*-value < 0.05]. We conclude that this low rate of TT vaccine uptake found in this study among pregnant women increases the number of newborn babies susceptible to contracting neonatal tetanus. The findings of this study should be considered in the development of communication strategies targeting and prioritizing at-risk groups to increase TT vaccine uptake among pregnant women in Sudan.

## 1. Introduction

Vaccination during pregnancy has been proven as an effective preventative measure to reduce the incidence of child and maternal morbidity and mortality worldwide. Vaccinating pregnant women and infants with the tetanus toxoid vaccine has been credited for the significant decrease in tetanus mortality by 87% (from 275,379 to 34,684) between 1990 and 2019 [1]. Despite this success, tetanus remains a main public health problem in many parts of the world where vaccination coverage is suboptimal, especially in low-income countries [2] where under-vaccination is compounded by unhygienic deliveries and abortions and poor postnatal hygiene and umbilical cord care practices [3,4,5,6,7,8]. As of 2017, maternal and neonatal tetanus is reported as a public health problem in 13 countries globally including Sudan [2,5].

Neonatal tetanus protection can be ensured when women either receive at least two doses of the tetanus toxoid vaccine, the last dose within the previous 3 years; receive at least three doses, the last within the previous 5 years; receive at least four doses, the last within the previous 10 years; or receive five or more doses anytime during their life [3]. According to WHO recommendations, women of reproductive age in countries where MNT is a public health issue should receive at least two doses of the TT vaccine with consideration of the previous doses during the childhood and adolescence period, giving a preference to a third dose. However, the WHO recommends that women of reproductive age should receive three doses of the TT vaccine, regardless of their previous vaccination status if they are in districts at high risk in countries that have not yet achieved the MNT elimination goal (i.e., less than 1 neonatal tetanus case per 1000 live births in every district) [3].

In Sudan, despite the high coverage of the first dose of the pentavalent (DTP-Hep-Hib) vaccine (92% in 2019) among the targeted infants in Sudan [9], the Simple Spatial Survey (S3M) reported in 2020 that only 67.34% of the women received two doses of the tetanus toxoid (TT) vaccine during their last pregnancy [10]. Additionally, neonatal tetanus was reported as one of the top 10 causes of death among children under 5 years in Sudan in 2017. One of the additional risk factors contributing to the maternal and neonatal tetanus (MNT) burden in Sudan includes the very low rate of postpartum care coverage (about 34.2%), where post-natal health checks for mothers and their newborns occur [10].

There are little data and reports on the reasons behind the low coverage of the tetanus toxoid vaccine among Sudanese women. However, an analysis of the MICS data shows that there is geographical inequality in the protection against MNT, as women (aged 15–49 years) who are residents in urban areas were more protected against neonatal tetanus (65.9%) than their counterparts in rural areas (55.4%). Moreover, protection against MNT is increased proportionally by the increase in levels of education among women of reproductive age. Similarly, differences were shown among women with varying economic status; the percentage of neonatal tetanus protection among women in the richest quintile was higher than those from the poorest quintile (74.2% and 44.4%, respectively) [11].

This study aimed to assess determinants of TT vaccine uptake among pregnant women in two public maternity specialized hospitals (Omdurman Maternity Hospital and Al Saudi Hospital) given the recommendation of the WHO for women who live in countries such as Sudan at high risk of MNT. The findings will be used to inform future programs and communication strategies to increase the coverage of maternal vaccination in Sudan.

## 2. Materials and Methods

### 2.1. Study Design

A hospital-based cross-sectional study was conducted at two public specialized hospitals (i.e., both provide obstetric and gynecological services), the Omdurman and Al Saudi maternity hospitals in Omdurman District, Khartoum State, in Sudan, from February to April 2020.

### 2.2. Population and Sampling

#### 2.2.1. Population

The study population included pregnant women aged 15–49 years residing in Sudan, who attended either of the two hospitals at the time of the study, regardless of their number of visits.

#### 2.2.2. Sampling and Sample Size

A total of 350 participants were recruited for the study using the following formula [12]:$$\left[n=\frac{z^{2}pq}{d^{2}}\;\right]$$
where
*n* = sample size*z* = (1 *− α*) is the z-score corresponding to a 95% confidence interval and was computed as 1.96.*p* = 0.288 which is the probability/percentage of women who received at least two doses of the TT vaccine during their last pregnancy in Khartoum State based on the 2014 MICS report.*q* = (1 − *p*) = 0.712*d* = desired margin of error of 0.05.

We selected proportionally 200 participants from Omdurman Maternity Hospital and 150 participants from Al Saudi Hospital according to the average number of patients who visit each hospital per month. Systematic random sampling was used to select pregnant women in this study until the estimated sample size was completed from each hospital.

### 2.3. Data Collection

Data were collected using a structured and pretested questionnaire. The interviews were conducted in the Arabic language. Data were collected using the pretested questionnaires obtained from pregnant women after their consultation with doctors. The dependent variable was uptake of the tetanus toxoid vaccine, which was measured as uptake of three doses and more (as recommended by the WHO for countries at high risk) and uptake of less than the recommended doses (i.e., two doses and fewer) at any point in time [3]. The independent variables included sociodemographic data, such as family income, which was measured as self-ranking, perceptions about tetanus and the tetanus vaccine, which were measured using a five-point Likert scale (strongly disagree/strongly agree), and reproductive health factors including the number and planning of pregnancy.

### 2.4. Statistical Analysis

Data analysis was performed using Statistical Package for Social Sciences (SPSS) software (Version 24). Pregnant women who do not know their TT vaccination status were excluded from the analysis. The chi-square test or Fisher’s exact test (when the count is <5 in a cell) was conducted to assess the factors associated with the uptake of the TT vaccine among pregnant women. A *p*-value of <0.05 was considered statistically significant. Variables that were significantly associated with the primary dependent variable were included in a logistic regression model to identify the predictors of TT vaccine uptake among pregnant women. We first calculated the odds ratios for each variable using univariate regression analysis and then we calculated the adjusted odds ratio using a multivariate logistic regression model.

### 2.5. Ethical Consideration

The study was approved by the Ahfad University for Women’s review board (IRB), and permission to enter the hospitals was obtained from the Khartoum State Ministry of Health and the hospitals’ general managers. Written informed consent was obtained from each of the participants.

## 3. Results

### 3.1. Characteristics of the Study Participants

As shown in Table 1, out of 313 participants, 173 (55.5%) attended Omdurman Maternity Hospital; the average age of these pregnant women was 28.33 years (SD = 6.310). Nearly half of the respondents, 142 (45.36%), had a university level of education followed by those who completed primary school, 94 (30%). Moreover, most of the respondents, 284 (90.7%), self-ranked their family income level as medium. One hundred and thirty-five of these pregnant women (42.8%) reported having one or two children (Table 1).

### 3.2. Perception of the Pregnant Women in Sudan towards the TT Vaccine

As shown in Figure 1, the majority of the participants strongly agreed (i.e., more confident) that the TT vaccine is important, safe, and effective against tetanus, and perceived tetanus as a serious disease (68.6%, 70.6%, 58%, and 70%).

### 3.3. Tetanus Vaccine Uptake

More than half of the pregnant women (51%) reported that they were partially vaccinated (1–2 doses) with the tetanus vaccine. Only 40% of these pregnant women were fully vaccinated with the tetanus vaccine (three and more doses). A very low proportion (8%) of these women reported that they have never been vaccinated with the tetanus vaccine (Figure 2).

### 3.4. Factors Associated with Tetanus Vaccine Uptake among Pregnant Women in Sudan

To understand the factors related to TT vaccine uptake among pregnant women, we ran the Chi-square test (and sometimes Fisher’s exact test when the counts in the cells are less than five). The results are summarized in Table 1. The percentage of pregnant women who received three or more doses of the vaccine (as recommended for women at high risk) at Al Saudi Hospital was less than those who attended Omdurman Maternity Hospital (*p* = 0.0120). In addition, most pregnant women who did not attend university were vaccinated with two or fewer doses of TT compared to those with a university degree of education. The highest percentage of pregnant women with fewer than the recommended doses were among those who have had 3–4 children. Uptake of the recommended dose of the TT vaccine was significantly associated with the hospital (*p* < 0.012), the mother’s education (*p* < 0.001), and the number of children (*p* < 0.001). On the other hand, there was no statistically significant difference in the uptake of the TT vaccine related to the family income or employment status of eth pregnant women.

### 3.5. Predictors of Tetanus Vaccine Uptake among Pregnant Women in Sudan

The predictors of tetanus vaccine uptake among pregnant women in Sudan are displayed in Table 2. The summary of the multivariate logistic regression analysis showed that pregnant women who attended Al Saudi Hospital were less likely to be vaccinated with the recommended dose (received ≥3 doses) compared to those who attended Omdurman Hospital [OR = 0.49 (95% C.I. 0.29–0.82), *p*-value < 0.05]. In addition, the number of children was a significant predictor of the mothers’ immunization status. Compared to pregnant women who have not had children yet, those with one to two children were three times more likely to have received the recommended dose of the TT vaccine [OR = 3.31 (95% C.I. 1.67–6.56), *p*-value < 0.05], while those with three to five children were five times more likely to receive ≥3 doses [OR = 5.05 (95% C.I. 2.37–10.79), *p*-value < 0.05], and those with five children or more were ten times more likely to have received ≥3 doses [OR = 10.54 (95% C.I. 4.30–25.86), *p*-value < 0.05].

## 4. Discussion

This study aimed to assess the determinants of tetanus toxoid (TT) vaccination among pregnant women at two public hospitals, Omdurman Maternity Hospital and Al Saudi Hospital in Khartoum State in 2020.

The findings of this study showed that the majority of the participants were very confident that the TT vaccine is important, safe, and effective and that the disease itself is serious (Figure 1). These findings are consistent with a global study conducted in over 140 countries in 2018, which found that people in the East Africa region are more likely to perceive the safety and effectiveness of vaccines (92% and 90%, respectively). Furthermore, the same report revealed that people in Ethiopia and Egypt are most likely to agree about the importance, safety, and effectiveness of vaccines [13].

According to the WHO recommendation, women of reproductive age should receive three doses of the TT vaccine, regardless of their previous vaccination status if they are in districts in countries at high risk that have not yet achieved MNT elimination status [3]. This study found that about 40% (Figure 1) of the pregnant women received three doses or more of the TT vaccine. This was lower than the result found from the Errer district, Somali Regional State, Eastern Ethiopia, and the proportion reported in Damboya Woreda, Kembata Tembaro Zone, SNNP, Ethiopia (51.8% and 72.5%, respectively [14,15]. This difference might be explained as a result of the very low rate of institutional deliveries in Sudan (about 27.7%); where post-natal health checks for the mothers and their newborns occur [10].

Interestingly, these study findings show that there are many factors increasing the probability of pregnant women being vaccinated or not against tetanus. We found that pregnant women who attended Al Saudi Hospital were less likely to be vaccinated against tetanus (mostly vaccinated with ≤2 doses). In addition, most pregnant women who did not attend university were not vaccinated against tetanus compared to those with a university degree of education. As reported in previous studies, the education level of pregnant women may affect their immunization status because it means they are likely to secure gainful employment thus guaranteeing the financial stability that enables easier access to better immunization services and healthcare in general [16]. Factors such as the negative perception of pregnant women about tetanus, poor planning of the upcoming pregnancy, and a higher number of pregnancies decrease the likelihood of vaccination against tetanus with a recommended dose. There was no statistically significant association between family income and the immunization status of pregnant women, though previous studies showed a significant association [17,18,19].

Social determinants have the potential to affect immunization programs around the world, especially factors related to support from the father/male spouse, disposable income available to the pregnant women, the general level of knowledge/understanding of the women regarding the tetanus vaccine, and the level of formal education all affect immunization and the uptake of vaccines. Exploring all these determinants affecting immunization is of great importance in the attempts to increase immunization uptake, reduce hesitancy to immunization and promote a culture that demands vaccines, and ultimately increase immunization rates, and prevent early infant mortality previously attributed to neonatal tetanus. Access to vaccines and vaccine equity across communities are key determinants of the control of vaccine-preventable diseases, and low-income countries require support from developed nations (both financial and logistical) if we are to reduce prevailing inequalities in vaccinations among different populations [19].

### Limitation

Due to some limitations in our study, its findings should be interpreted within its context. Since this was a cross-sectional survey, it is difficult to explain if the factors described cause a lack of protection from the tetanus vaccine, but the significant factors do explain the factors underlying low vaccination. Moreover, the results of the current study are prone to recall bias, as the participants did not show their vaccination cards during data collection. Nonetheless, the study provided valuable information that could be used in developing health education interventions to increase TT vaccine uptake among pregnant women in Sudan.

## 5. Conclusions

We conclude that the low rate of TT vaccine uptake found in this study among pregnant women increases the number of newborn babies susceptible to contracting neonatal tetanus. This study provides useful information that can inform the development of communication strategies targeting and prioritizing at-risk groups to increase TT vaccine uptake among pregnant women in Sudan. Additionally, it can help in developing policies to eliminate MTN in Sudan.

## Figures and Tables

**Figure 1 vaccines-11-01268-f001:**
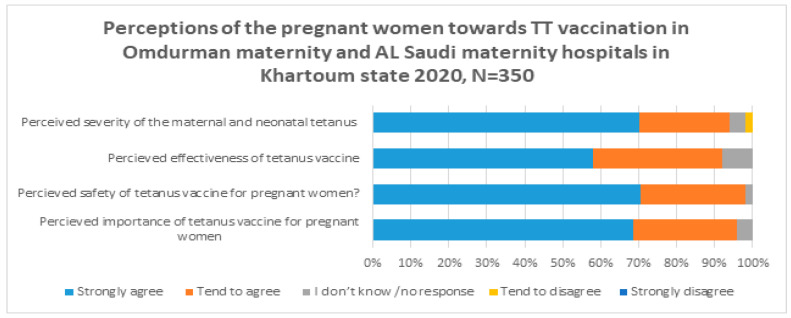
Perceptions of pregnant women towards TT vaccination at Omdurman Maternity Hospital and Al Saudi Hospitals in Sudan.

**Figure 2 vaccines-11-01268-f002:**
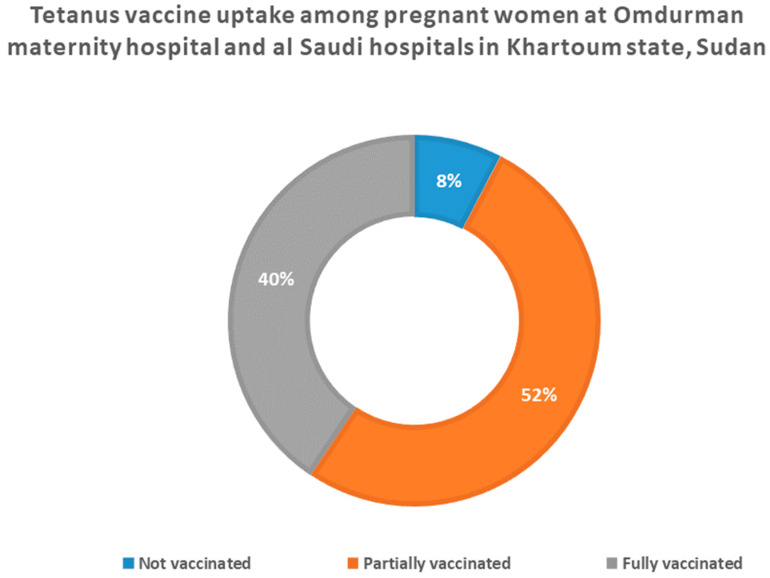
Tetanus vaccine uptake among pregnant women at Omdurman Maternity and Al Saudi Hospitals in Khartoum state, Sudan.

**Table 1 vaccines-11-01268-t001:** Participants’ profile and factors associated with tetanus vaccine uptake among pregnant women at Omdurman Maternity Hospital and Al Saudi Hospital in Sudan (N = 313).

Participants’ Profile		TT Immunization Status			
		Fewer than the Recommended Doses (≤2 Doses) N (%) = 186 (59.4%)	Recommended Doses (≥3 Doses) N (%) = 127 (40.6%)	Total N (%)	*p*-Value
Hospital	Omdurman Maternity Hospital	92 (53.2%)	81 (46.8%)	173 (55.2%)	0.012 *
	Al Saudi Hospital	94 (67.1%)	46 (32.9%)	140 (44.7%)	
Mother’s Education	Not educated	4 (66.7%)	2 (33.3%)	6 (1.9%)	0.001 * ^a^
	Primary	68 (72.3%)	26 (27.7%)	94 (30%)	
	Secondary	49 (69.0%)	22 (31.0%)	71 (22.68%)	
	University	65 (45.8%)	77 (54.2%)	142 (45.36%)	
Family income	High	9 (69.2%)	4 (30.8%)	13 (4.15%)	0.134
	Medium	164 (57.7%)	120 (42.3%)	284 (90.7%)	
	Low	13 (81.3%)	3 (18.8%)	16 (5.11%)	
Number of children	1–2 children	86 (64.2%)	48 (35.8%)	134 (42.8%)	0.001* ^a^
	3–4 children	63 (72.4%)	24 (27.6%)	87 (27.79%)	
	5 children and more	37 (40.2%)	55 (59.8%)	92 (29.39%)	
Mother’s employment	Not employed	155 (58.3%)	111 (41.7%)	266 (85.0%)	0.322
	Employed	31 (66.0%)	16 (34.0%)	47 (15.0%)	

* Statistically significant; ^a^ Fisher’s exact test.

**Table 2 vaccines-11-01268-t002:** Predictors of tetanus vaccine uptake among pregnant women in Sudan (N = 313).

Predictors	OR (95% C.I. of OR)	aOR (95% C.I. of aOR)
**Hospital**		
Omdurman Maternity Hospital		
Al Saudi Hospital	0.56 (0.350–0.882) *	0.49 (0.29–0.82) *
**Mother’s Education**		
Not educated ^®^		
Primary	0.77 (0.13–4.43)	0.5 (0.07–3.51)
Secondary	0.89 (0.15–5.27)	0.62 (0.09–4.72)
University	2.37 (0.42–13.35)	2.05 (0.30–13.99)
**Number of Children**		
Have not had children yet ^®^		
1–2 children	2.9 (1.522–5.520) *	3.31 (1.67–6.56) *
3–4 children	4.77 (2.34–9.75) *	5.05 (2.37–10.79) *
5 children and more	5.11 (2.26–11.54) *	10.54 (4.30–25.86) *

* Statistically significant at *p*-value < 0.05; aOR = adjusted odds ratio; ^®^ = reference category.

## Data Availability

The data presented in this study are available from the corresponding author upon request.

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
