# Peer review of "Predictors of Tetanus Vaccine Uptake among Pregnant Women in Khartoum State, Sudan: A Hospital-Based Cross-Sectional Study"

_vaccines, 2023, doi:10.3390/vaccines11071268_

Round 1
Reviewer 1 Report
This paper describes a cross-sectional survey investigating factors related to maternal tetanus toxoid vaccination conducted in 2 hospitls in Sudan. This is an important topic and the results potentially identify high-risk groups where vaccination should be targeted.
However, in my opinion, the paper requires some additional clarifications to be able to fully understand the authors conclusions, particularly around the definition of protection from tetanus (the main outcome) as they rely purely on vaccination recall and the definition of ‘protection’ is linked to recall of vaccination doses over an unclear period of time.
My specific comments are below.
Introduction
Line 44: Citation [1] is not relevant – please cite the original source of data mentioned in the sentence
Line 46: Similarly reference [2] is from 2010 and therefore questionable whether it is suitable to support the statement about current vaccination coverage (or latterly where 2017 MNT data are included) (Line 49)
Line 50 – please give relevant data i.e 2017 data if linking to 2014 cluster survey or vice versa.
Introduction should include a clear description of the link between vaccination, vaccination recall and actual immunological protection.
Methods:
Population: were women attending for 1st or subsequent visits to these hospitals (for the current pregnancy)?
Sample size calculation: the methodology and formula is not needed, however a clearer explanation of why the 0.288 value was chosen as being the probability of receiving at least 2 doses. How is this relevant – what were the comparison groups? If this is a qualitative survey and there are pragmatic/saturation considerations, then state these.
Data collection: state how those administering the questionnaires were trained and their role (i.e. clinic staff, study staff etc). State clearly how protection is defined (i.e. 2 doses of TT ) and within how long? What about those who may have had a full course before pregnancy etc? Was any secondary verification available (e.g. vaccination records)?
Why is ‘not protected’ including ≤2 doses when WHO’s aim has been to provide at least two TT containing vaccines during pregnancy? TT2 is one of the primary metrics of the MNT programme, so excluding those with 2 doses needs a clear explanation. [See end of this document for WHO recommendations] In the abstract, protection appears to be defined as “received 3 doses or more (protected) of TT vaccine during their current pregnancy” – however WHO’s recommendation for is for “at least 2 doses of TTCV, preferably Td, with an interval of at least 4 weeks between doses and the second dose at least 2 weeks before the birth. To ensure protection for a minimum of 5 years, a third dose should be given at least 6 months later.” Is this achievable within a single pregnancy, as implied by the authors?
Give a clear description of how household incomes were categorised.
Statistical analysis should give more details about methodology of logistic regression and adjustment for odds ratios.
Results:
Include a flow chart of recruitment and the numbers excluded for not included, including details of those consented but excluded (and reasons for this).
Include in the table the stage of pregnancy women were attending at (i.e. trimester)
Much of the text is repeating the tables and could be reduced.
Discussion
This section should mention the major limitations of this study:
1. Recall as a poor predictor of vaccination history
2. That vaccination does not mean protection and the timing of previous doses should be considered.
3. That for many years and in most countries 2 doses of TT containing vaccine for pregnant women has been considered sufficient for tetanus protection.
4. The exclusion of those who could not remember
For additional reference WHO position paper on tetanus guidance is here: https://www.who.int/publications/i/item/WHO-WER9206
“Pregnant women and their newborn infants are protected from birth-associated tetanus if the mother received either 6 TTCV doses during childhood or doses if first vaccinated during adolescence/adulthood (documented by card, immunization registry and/or history) before the time of reproductive age. Vaccination history should be verified in order to determine whether a dose of TTCV is needed in the current pregnancy.
In countries where MNT remains a public health problem, pregnant women for whom reliable information on previous tetanus vaccinations is not available should receive at least 2 doses of TTCV, preferably Td, with an interval of at least 4 weeks between doses and the second dose at least 2 weeks before the birth. To ensure protection for a minimum of 5 years, a third dose should be given at least 6 months later. A fourth and fifth dose should be given at intervals of at least 1 year, or in subsequent pregnancies, in order to ensure lifelong protection.
Pregnant women who have received only 3 doses of TTCV during childhood without booster doses should receive 2 doses of TTCV at the earliest opportunity during pregnancy with a minimal interval of 4 weeks between doses and the second dose at least 2 weeks before giving birth. Although 1 booster dose should result in a rapid increase in antibody, the level of tetanus-specific antibodies in women who received only a 3-dose primary series during infancy is similar to that of unimmunized individuals 15 years post-immunization. Therefore, 2 doses are recommended in order to ensure a total of 5 doses before delivery. Women who received 4 TTCV doses during childhood or pre-adulthood need only 1 booster dose, which should be given at the first opportunity. In both scenarios, to provide lifelong protection, a sixth dose would be needed at least 1 year after the fifth dose.”
Author Response
However, in my opinion, the paper requires some additional clarifications to be able to fully understand the authors conclusions, particularly around the definition of protection from tetanus (the main outcome) as they rely purely on vaccination recall and the definition of ‘protection’ is linked to recall of vaccination doses over an unclear period of time.
My specific comments are below.
Introduction
Line 44: Citation [1] is not relevant – please cite the original source of data mentioned in the sentence
Answer: Revised
Line 46: Similarly reference [2] is from 2010 and therefore questionable whether it is suitable to support the statement about current vaccination coverage (or latterly where 2017 MNT data are included) (Line 49)
Answer: Revised
Line 50 – please give relevant data i.e 2017 data if linking to 2014 cluster survey or vice versa.
Answer: Revised and we added recent data about vaccination coverage in 2019 and 2020.
Introduction should include a clear description of the link between vaccination, vaccination recall and actual immunological protection.
Answer: Revised as suggested. As this is a short report, we added a short description of the recommendation of WHO regarding the number of doses: ‘’According to WHO recommendations, women of reproductive age in countries where MNT is a public health issue should receive at least two doses of the TT vaccine, with giving a preference to a third dose. However, WHO recommends that women of reproductive age should receive 3 doses of TT vaccine, regardless of their previous vaccination status if they are in high at risk districts in countries that have not yet achieved MNT elimination goal (i.e. less than 1 neonatal tetanus case per 1000 live births in every district).’’
Methods:
Population: were women attending for 1st or subsequent visits to these hospitals (for the current pregnancy)?
Answer: We included pregnant women aged 15–49 years who attended either of the two hospitals at the time of the study, ‘’regardless of their number of visits’’.
Sample size calculation: the methodology and formula is not needed, however a clearer explanation of why the 0.288 value was chosen as being the probability of receiving at least 2 doses. How is this relevant – what were the comparison groups? If this is a qualitative survey and there are pragmatic/saturation considerations, then state these.
Answer: This study is qualitative survey as we indicated in the study design ‘’A hospital-based cross-sectional study’’ to address the purpose of the study ‘’Findings will be used to inform future programs and communication strategies to increase coverage of maternal vaccination in Sudan.’’
Data collection: state how those administering the questionnaires were trained and their role (i.e. clinic staff, study staff etc). State clearly how protection is defined (i.e. 2 doses of TT ) and within how long? What about those who may have had a full course before pregnancy etc? Was any secondary verification available (e.g. vaccination records)?
Answer: There was no secondary verification available at the time of the data collection. We added this issue to the limitation section.
Why is ‘not protected’ including ≤2 doses when WHO’s aim has been to provide at least two TT containing vaccines during pregnancy? TT2 is one of the primary metrics of the MNT programme, so excluding those with 2 doses needs a clear explanation. [See end of this document for WHO recommendations] In the abstract, protection appears to be defined as “received 3 doses or more (protected) of TT vaccine during their current pregnancy” – however WHO’s recommendation for is for “at least 2 doses of TTCV, preferably Td, with an interval of at least 4 weeks between doses and the second dose at least 2 weeks before the birth. To ensure protection for a minimum of 5 years, a third dose should be given at least 6 months later.” Is this achievable within a single pregnancy, as implied by the authors?
Answer: Thanks for your recommendation. We revised the scope of the manuscript which is now amended from the protected dose to the recommended dose for ‘’at high risk’’ countries as recommended by WHO (Reference No. 3).
Answer: We argue that the same recommendation emphasized that ‘In countries which have not achieved MNTE status (<1 neonatal tetanus case per 1000 live births in every district), the “high-risk” approach should be part of the elimination strategy. This approach targets all WRA in high-risk districts and consists of 3 campaign[1]style vaccination rounds to provide 3 doses of TTCV irrespective of previous vaccination status, with an interval of at least 4 weeks between doses 1 and 2, and at least 6 months between doses 2 and 3’’. Sudan is one of the countries that does not achieve MNT Elimination status due to several factors that were mentioned in the introduction.
Give a clear description of how household incomes were categorised.
Answer: We described the family income ‘which was measured as self-ranking’’
Statistical analysis should give more details about methodology of logistic regression and adjustment for odds ratios.
Answer: we added more details as you recommended regarding the methodology of logistic regression in the sub- section of the statistical analysis: ‘We firstly calculated the odds ratios for each variable using univariate regression analysis and then we calculated the adjusted odds ratio using a multivariate logistic regression model’
Results:
Include a flow chart of recruitment and the numbers excluded for not included, including details of those consented but excluded (and reasons for this).
Include in the table the stage of pregnancy women were attending at (i.e. trimester)
Answer: We did not ask a question about the stage of pregnancy because the scope of the study to find the association between the social and behavioural factors with the tetanus vaccine uptake.
Much of the text is repeating the tables and could be reduced.
Answer: Revised
Discussion
This section should mention the major limitations of this study:
- Recall as a poor predictor of vaccination history
- That vaccination does not mean protection and the timing of previous doses should be considered.
- That for many years and in most countries 2 doses of TT containing vaccine for pregnant women has been considered sufficient for tetanus protection.
- The exclusion of those who could not remember
For additional reference WHO position paper on tetanus guidance is here: https://www.who.int/publications/i/item/WHO-WER9206
Answer: Thanks for sharing this very important reference which we already used in the manuscript (Reference No. 3). As we already amended the manuscript substantially and shifted the scope from the protected doses to the recommended doses for high-risk countries, we added additional limitation regarding the recall bias ‘’Moreover, the results of the current study are prone to recall bias, as the participants did not show their vaccination cards during the data collection’’.
Reviewer 2 Report
"Predictors of Tetanus Vaccine Uptake Among Pregnant Women in Sudan: A Hospital-Based Cross-Sectional Study"
General Comments:
The article presents a hospital-based cross-sectional study aimed to assess the predictors of tetanus toxoid (TT) vaccine uptake among pregnant women in two public maternity specialized hospitals in Sudan. The study recruited 350 pregnant women, with 313 participants included in the analysis. The study found that only 40% of the pregnant women received 3 doses or more (protected) of TT vaccine during their current pregnancy. Pregnant women who attended Al Saudi hospital were less likely to be protected against Tetanus compared to those who attended Omdurman hospital. Furthermore, the number of children at home was a significant predictor of mothers’ immunization status, as those with five children or more were ten times more likely to be protected against tetanus.
The study has several strengths, including the large sample size, the use of a pretested structured questionnaire, and the multivariate analysis performed to identify the predictors of TT vaccine uptake. However, there are some limitations to the study that should be addressed. For example, the study only included pregnant women attending two hospitals, which may not be representative of the general population of pregnant women in Sudan. Additionally, the study relied on self-reported data, which may be subject to recall bias or social desirability bias. Lastly, the study did not explore the reasons why some pregnant women did not receive the TT vaccine, which could provide valuable insights into how to increase vaccine uptake.
Overall, the study provides useful information that can inform the development of communication strategies targeting and prioritizing at-risk groups to increase TT vaccine uptake among pregnant women in Sudan.
Specific Comments:
Title:
The title is clear and concise and accurately reflects the content of the article. However, the term Sudan used could denote the whole population was tested-suggest to change to Omdurman district, Khartoum state in Sudan
Abstract:
The abstract provides a clear and concise summary of the study, including the background, methods, results, and conclusions. However, it would be helpful to include the main findings in the abstract to provide readers with a quick overview of the study.
Introduction:
The introduction provides a good overview of the background and rationale for the study. However, it would be helpful to include a brief discussion of the current state of tetanus vaccine uptake among pregnant women in Sudan and the potential consequences of low vaccine uptake.
Methods:
The methods section is well-written and provides a clear and concise description of the study design, sampling, data collection, and statistical analysis. However, it would be helpful to include more information on the recruitment process and the response rate to provide readers with a better understanding of the representativeness of the study sample.
Results:
The results section presents the study findings clearly and concisely. The use of tables and figures is helpful in summarizing the data. However, it would be helpful to include confidence intervals for the proportions presented in the tables to provide readers with a sense of the precision of the estimates.
Conclusion:
The conclusion is clear and concise and accurately reflects the study findings. However, it would be helpful to include a brief discussion of the implications of the study for policy and practice to provide readers with a sense of how the study findings can be translated into action.
Author Response
Reviewer 2
"Predictors of Tetanus Vaccine Uptake Among Pregnant Women in Sudan: A Hospital-Based Cross-Sectional Study"
General Comments:
The article presents a hospital-based cross-sectional study aimed to assess the predictors of tetanus toxoid (TT) vaccine uptake among pregnant women in two public maternity specialized hospitals in Sudan. The study recruited 350 pregnant women, with 313 participants included in the analysis. The study found that only 40% of the pregnant women received 3 doses or more (protected) of TT vaccine during their current pregnancy. Pregnant women who attended Al Saudi hospital were less likely to be protected against Tetanus compared to those who attended Omdurman hospital. Furthermore, the number of children at home was a significant predictor of mothers’ immunization status, as those with five children or more were ten times more likely to be protected against tetanus.
The study has several strengths, including the large sample size, the use of a pretested structured questionnaire, and the multivariate analysis performed to identify the predictors of TT vaccine uptake. However, there are some limitations to the study that should be addressed. For example, the study only included pregnant women attending two hospitals, which may not be representative of the general population of pregnant women in Sudan. Additionally, the study relied on self-reported data, which may be subject to recall bias or social desirability bias. Lastly, the study did not explore the reasons why some pregnant women did not receive the TT vaccine, which could provide valuable insights into how to increase vaccine uptake.
Overall, the study provides useful information that can inform the development of communication strategies targeting and prioritizing at-risk groups to increase TT vaccine uptake among pregnant women in Sudan.
Answer: The majority of the clients of these maternity hospitals are pregnant women either coming for antenatal care visit or for delivery. We revised the limitation and added recall bias as one of the limitation.
Specific Comments:
Title:
The title is clear and concise and accurately reflects the content of the article. However, the term Sudan used could denote the whole population was tested-suggest to change to Omdurman district, Khartoum state in Sudan
Answer: Revised and as you suggested, but we added Khartoum state, as the two hospitals are the main hospitals for maternity in Sudan.
Abstract:
The abstract provides a clear and concise summary of the study, including the background, methods, results, and conclusions. However, it would be helpful to include the main findings in the abstract to provide readers with a quick overview of the study.
Answer: Revised
Introduction:
The introduction provides a good overview of the background and rationale for the study. However, it would be helpful to include a brief discussion of the current state of tetanus vaccine uptake among pregnant women in Sudan and the potential consequences of low vaccine uptake.
Answer: We added a description about the TT vaccine uptake in Sudan as you suggested.
Methods:
The methods section is well-written and provides a clear and concise description of the study design, sampling, data collection, and statistical analysis. However, it would be helpful to include more information on the recruitment process and the response rate to provide readers with a better understanding of the representativeness of the study sample.
Answer: We did not include response rate in the study as every pregnant woman meeting the criteria of inclusion was included in the study until the required sample size was achieved from each hospital.
Results:
The results section presents the study findings clearly and concisely. The use of tables and figures is helpful in summarizing the data. However, it would be helpful to include confidence intervals for the proportions presented in the tables to provide readers with a sense of the precision of the estimates.
Answer: As this study is focusing mainly on the predictors of TT vaccine , we already added Confidence Intervals for the Odds Ratios (Adjusted and non-adjusted) to present the precision of the estimates.
Conclusion:
The conclusion is clear and concise and accurately reflects the study findings. However, it would be helpful to include a brief discussion of the implications of the study for policy and practice to provide readers with a sense of how the study findings can be translated into action.
Answer: We added the implications of the study for policy and practice in the conclusion
Reviewer 3 Report
Several minor comments:
Line 33 in Abstract: English can be improved.
Line 113: English can be improved
Line 131: 1 decimal place is only required
Line 233 English unclear
Figure 1: Are major differences revealed when this Figure is split by high/medium/low income?
A comprehensive literature review of all studies relating to the determinants of Tetanus vaccination should be included.
Suggestions regarding how the situation can be improved should be discussed.
Your study design should have included feedback on barriers to vaccination and hoe coverage could be improved.
Author Response
Review 3
Several minor comments:
Line 33 in Abstract: English can be improved.
Answer: Revised as you suggested.
Line 113: English can be improved
Answer: Revised as you suggested.
Line 131: 1 decimal place is only required
Answer: Revised as you suggested.
Line 233 English unclear
Answer: Revised as you suggested.
Figure 1: Are major differences revealed when this Figure is split by high/medium/low income?
Answer: Although the variables in the figures are independent variables in this study, however, we think there may not be any differences.
A comprehensive literature review of all studies relating to the determinants of Tetanus vaccination should be included.
Answer: Despite the paucity of literature in Sudan, wee added some evidence from Sudan showing the rate of vaccination uptake, factors associated and WHO recommendation regarding the number of doses. as you suggested.
Suggestions regarding how the situation can be improved should be discussed.
Answer: We added some suggestions to improve the situation in the conclusion as you suggested.
Your study design should have included feedback on barriers to vaccination and hoe coverage could be improved.
Answer: We added some suggestions to improve the situation in the conclusion as you suggested
Round 2
Reviewer 1 Report
Thank you for the changes. The review reads much better now and I think the emphasis that this is a high risk area and that 3 TT doses should be given enhances the paper.
However, this focuses on now the fact that 3 TT doses are given during the CURRENT pregnancy - is this correct? If this is so then in the discussion it must be stated clearly that it is possible (and maybe an estimate of how possible) it is that women have had doses BEFORE the current pregnancy. Indeed is this not actually the case - i.e. why would the authors expect the women who have had multiple pregnancies to have more doses of TT in THIS pregnancy.
I also suggest shortening the sample size section as previously.
Author Response
Response: Thank you very much for your critical revision. Your perspective has greatly improved our manuscript.
Thank you for the changes. The review reads much better now and I think the emphasis that this is a high risk area and that 3 TT doses should be given enhances the paper.
However, this focuses on now the fact that 3 TT doses are given during the CURRENT pregnancy - is this correct? If this is so then in the discussion it must be stated clearly that it is possible (and maybe an estimate of how possible) it is that women have had doses BEFORE the current pregnancy. Indeed is this not actually the case - i.e. why would the authors expect the women who have had multiple pregnancies to have more doses of TT in THIS pregnancy.
Response: Your point is valid. However, when we asked the pregnant women about their tetanus vaccination status, we did not specify the current pregnancy. As we supposed that a period of one pregnancy is sufficient for the three doses of tetanus vaccination to be taken (with an interval of at least 4 weeks between doses 1 and 2, and at least 6 months between doses 2 and 3 as recommended by WHO). Our findings support our hypothesis, as the number of children is associated with the number of doses of tetanus vaccine.
I also suggest shortening the sample size section as previously.
Response: For educational purposes, we kept the equation, but we shortened the description of sample size and sampling.
Reviewer 3 Report
Changes have been made as requested.
Author Response
Thank you very much for your critical revision. Your perspective has greatly improved our manuscript
Round 3
Reviewer 1 Report
Thank you for the changes and clarification. With this in mind, there are still some modifications which are needed as in the author's response it appears that the questions concerned TT vaccination applies to vaccination recall regarding any TT vaccination at any point in the past (which is perfectly reasonable). With this in mind, it needs to be very clearly written throughout the article about this.
Abstract: Lines 26 27 – remove ‘during their current pregnancy’ as this is incorrect
Methods
1. Lines 116-118 – clarify if this is at any point (i.e. not necessarily in the current pregnancy)
Discussion
1. Lines 209 and 210 – this is incorrect – remove the ‘during this pregnancy’
Author Response
Reviewer 2 comments
Thank you for the changes and clarification. With this in mind, there are still some modifications which are needed as in the author's response it appears that the questions concerned TT vaccination applies to vaccination recall regarding any TT vaccination at any point in the past (which is perfectly reasonable). With this in mind, it needs to be very clearly written throughout the article about this.
Response: Thank you for your constructive guidance. We are happy to make the suggestions as requested.
Abstract: Lines 26 27 – remove ‘during their current pregnancy’ as this is incorrect
Response: Revised.
Methods
- Lines 116-118 – clarify if this is at any point (i.e. not necessarily in the current pregnancy)
Response: Revised.
Discussion
- Lines 209 and 210 – this is incorrect – remove the ‘during this pregnancy’
Response: Revised.